# Therapeutic Efficacy of Adipose Tissue-Derived Components in Neuropathic Pain: A Systematic Review

**DOI:** 10.3390/bioengineering11100992

**Published:** 2024-09-30

**Authors:** Anouk A. E. Claessens, Linda Vriend, Zachri N. Ovadja, Martin C. Harmsen, Joris. A. van Dongen, J. Henk Coert

**Affiliations:** 1Department of Plastic, Reconstructive and Hand Surgery, Medisch Centrum Leeuwarden, 8934 AD Leeuwarden, The Netherlands; znovadja@gmail.com; 2Department of Plastic, Reconstructive and Hand Surgery, University Medical Center Utrecht (UMC Utrecht), 3584 CX Utrecht, The Netherlands; lindavriend@gmail.com (L.V.); jorisavandongen@gmail.com (J.A.v.D.); j.h.coert@umcutrecht.nl (J.H.C.); 3Department of Pathology and Medical Biology, University Medical Center Groningen, University of Groningen, 9713 GZ Groningen, The Netherlands; m.c.harmsen@rug.nl

**Keywords:** neuropathic pain, adipose stromal cells, stromal vascular fraction, autologous fat grafting

## Abstract

Background: Neuropathic pain results from a defect in the somatosensory nervous system caused by a diversity of etiologies. The effect of current treat-ment with analgesics and surgery is limited. Studies report the therapeutic use of adipose tissue-derived components to treat neuropathic pain as a new treatment modality. Objective: The aim of this systematic review was to investigate the therapeutic clinical efficacy of adipose tissue-derived components on neuro-pathic pain. Methods: PubMed, Medline, Cochrane and Embase databases were searched until August 2023. Clinical studies assessing neuropathic pain after autologous fat grafting or the therapeutic use of adipose tissue-derived com-ponents were included. The outcomes of interest were neuropathic pain and quality of life. Results: In total, 433 studies were identified, of which 109 dupli-cates were removed, 324 abstracts were screened and 314 articles were excluded. In total, ten studies were included for comparison. Fat grafting and cellular stromal vascular fraction were used as treatments. Fat grafting indications were post-mastectomy pain syndrome, neuromas, post-herpetic neuropathy, neuro-pathic scar pain and trigeminal neuropathic pain. In seven studies, neuropathic pain levels decreased, and overall, quality of life did not improve. Conclusions: The therapeutic efficacy of adipose tissue-derived components in the treatment of neuropathic pain remains unclear due to the few performed clinical trials with small sample sizes for various indications. Larger and properly designed (randomized) controlled trials are required.

## 1. Introduction

Neuropathic pain is a complex condition that results from a defect or disease in the somatosensory nervous system. It is caused either by intrinsic peripheral nerve problems, such as gene defects in sensory neurons or by the injury of peripheral nerves or the central nervous system [1,2]. Neuropathic pain manifests as burning, electric or pricking paroxysmal pain and is often paired with dysesthesia, paresthesia, allodynia and hyperalgesia [3]. Together, these symptoms greatly impair a patient’s quality of life (QoL) and form a significant societal healthcare burden paired with high costs [4].

To date, multiple treatment modalities exist to treat neuropathic pain. First-line treatment consists of (combined) analgesic drug therapy with tricyclic antidepressants (TCAs), serotonin or noradrenaline reuptake inhibitors (SNRIs) and anti-epileptics. The efficacy of these drugs is limited and often accompanied by adverse side effects. In refractory cases, neuromodulation (spinal cord or peripheral nerve) or neurosurgical intervention with resection and the relocation of the nerve can be performed, although there is little evidence for its efficacy [2]. 

In the literature, the potential of adipose tissue-derived components to treat neuropathic pain has gained interest in the last decade. Adipose tissue comprises adipocytes and the stromal vascular fraction (SVF) con-taining hematopoietic cells, adipose-derived stromal cells (ASCs), fibro-blasts, endothelial cells and connective tissue [5]. ASCs are progenitor cells attached around the vessels as precursor cell types, e.g., pericytes and supra-adventitial cells that support immunomodulation, remye-lination and axonal growth [6,7]. Adipose tissue is easily harvested from the abdomen due to its accessibility and the minimally invasive technique required to harvest it [8]. The process to culture and expand ASCs for clinical use is expensive and time-consuming. In addition, the therapeutic use of ASCs is subject to strict laws and regulations, although a few clinical trials have been registered [9]. Moreover, the retention and, therefore, anticipated therapeutic efficacy of injected ASCs and their secretome are low, as rapid diffusion in tissue is expected without a carri-er. Thus, the injection of mechanically isolated SVFs with a persevered ex-tracellular matrix as a carrier or lipografting will likely be more effective than the injection of cultured ASCs. 

The clinical application of adipose tissue has been mostly described in chronicchronic wounds and soft tissue reconstructions [10]. Little is known about the role of adipose tissue in the treatment of neuropathic pain [9,11,12]. This systematic review, therefore, aims to investigate the therapeutic clinical efficacy of adipose tissue-derived components in neuropathic pain. 

## 2. Methods

### 2.1. Protocol and Registration

This article follows the Preferred Reporting Items for Systematic Reviews and Meta-analysis (PRISMA) statement [13]. This study was not registered. 

### 2.2. Search Strategy

A systematic literature search was performed in the electronic databases PubMed, Medline, Cochrane and Embase from inception to August 2023. This search strategy, based on the PICO (The Patient, Intervention, Comparison, Outcome) framework, combined the following terms: “neuropathic pain” or “neuropathy” or “neuralgia” or “peripheral neuropathy” or “peripheral neuropathic pain” or “polyneuropathy” with “stromal vascular fraction” or “autologous fat grafting” or “fat grafting” or “allogeneic fat graft” or “adipose derived stem cells” (Appendix A). 

### 2.3. Eligibility Criteria

Studies were considered eligible if the efficacy of adipose tissue-derived components, i.e., adipose tissue, lipoaspirate, SVF or ASCs, was therapeutically tested for neuropathic pain in a clinical study. The exclusion criteria included no availability of the full-text article or English translation, (systematic) reviews, case reports, case series, letters to the editor, conference abstracts and animal and in vitro studies. 

### 2.4. Study Selection and Data Collection

Two reviewers (A.C. and Z.N.O.) independently assessed the titles, abstracts and full texts. Disagreements were resolved through discussions between the two reviewers. 

### 2.5. Data Extraction

Full-text articles were extracted and then graded following the MINOR criteria (Methodological Index for Non-Randomized Studies) [14]. Generally, study characteristics, treatment characteristics and neuropathic pain outcomes were extracted for each study. For adipose tissue-derived therapy, outcomes of interest included the type of adipose tissue-derived component, harvesting and processing techniques, injected volumes, dose frequency and supplementation and isolation procedures (if applicable). For neuropathic pain, outcomes of interest were changes in pain experience and indications of neuropathic pain. Differences were reported when there was a statistically significant difference (*p* < 0.05).

### 2.6. Risk of Bias in Individual Studies

Detailed information was provided for each included study to assess the risk of bias for possible different interventions, outcome measures and a broad spectrum of measurements. The Methodology Index for Non-Randomized Studies (MINORS) was used to determine the quality of non-randomized surgical studies [14]. 

## 3. Results

### 3.1. Study Selection

In total, 433 studies were identified. After the removal of duplicates, 326 abstracts were screened. In total, 10 studies were included, and 316 articles were excluded for several reasons (Figure 1). The MINORS criteria for each included study are mentioned in Table 1. 

### 3.2. Study Characteristics

The included studies were published between 2011 and 2022 (Table 2). A total of 461 participants were described in the included studies (range: 5 to 209 per study). The mean age was 54.7 years (range: 17 to 94 years), and 93.5% of all patients were female. Six studies had a female sex bias due to the inclusion of female participants only [17,18,20,21,23,24]. The indications for adipose tissue-derived treatment were post-mastectomy pain syndrome (PMPS) [17,18,20,21,23], neuropathic pain from neuromas [18], post-herpetic neuropathy [22], neuropathic scar pain [18], trigeminal neuropathic pain [24] and neuropathic pain without a specific cause [14]. Table 3 and Table 4 show an overview of the corresponding methods of harvesting, (laboratory) processing and injecting techniques for autologous fat and SVF-enriched fat. Table 5 shows per clinical trial its study population and corresponding results of the treatment of neuropathic pain.

### 3.3. Included Studies

#### 3.3.1. Post-Mastectomy Pain Syndrome

Caviggioli et al. performed two studies to investigate the efficacy of autologous fat grafting (AFG) in PMPS patients. The first study included patients with severe scar retractions (n = 113 from which15 lost to follow-up), while the second study included patients with severe scar retractions and radio-dystrophy (n = 209 from which 19 lost to follow-up) [17,18]. In the first study, 28 out of 34 patients stopped their analgesic drug therapy in the treatment group [17]. In the second study, 48 patients stopped their analgesic drug therapy in the treatment group. In both studies, it was not reported if patients stopped analgesic drug therapy in the control group (Table 5) [18].

Juhl et al. (2016) studied the efficacy of AFG in patients with unilateral PMPS (n = 15) in a randomized, controlled trial (RCT) [20]. Improved health-related QoL scores (DoloTest) were shown in the treatment group after 3 and 6 months compared to baseline scores. The control group remained unchanged compared to baseline scores. Treatment outcomes were not compared to the control outcomes. 

Lisa et al. (2020) performed a non-controlled, prospective multicenter trial to study the efficacy of AFG in PMPS patients (n = 37) [21]. VAS scores, as well as patient and observer POSAS scores, were reduced up to 6 months after treatment.

Most recently, Sollie et al. (2022) performed a single-center prospective, double-blind RCT to study the efficacy of AFG combined with scar release in patients with PMPS (n = 37, 2 lost to follow-up). There were no differences in the average and maximum pain levels, NPSI nor the SF-36 scores in the treatment group (n = 18) compared to the control group (n = 17) [23].

#### 3.3.2. Euromas

Calcagni et al. (2016) assessed the efficacy of SVF-enriched AFG after the microsurgical resection of end-neuromas to areas of painful end-neuromas in five patients in a retrospective study design without a control group. No differences in several symptoms (including spontaneous pain, spikes, hyperesthesia, tap pain and motion pain) were observed [16].

#### 3.3.3. Post-Herpetic Neuralgia

Sollie et al. (2020) performed a non-controlled, prospective single-center study to assess the efficacy of AFG in patients with post-herpetic neuralgia (n = 10) [22]. VAS and NPSI scores decreased after treatment, while SF-36 scores increased after surgery (*p* = 0.40) [22].

#### 3.3.4. Neuropathic Scar Pain

Huang et al. (2015) studied the efficacy of AFG in patients with painful neuropathic scars (n = 13). There was no control group. The mean VAS and NPSI scores were taken after treatment, and only one patient received pharmacologic therapy for pain control since no considerable improvement was experienced. This patient stopped pharmacologic therapy 1 month after their second AFG treatment [19].

#### 3.3.5. Trigeminal Neuropathic Pain

Vickers et al. (2014) studied the efficacy of cellular SVF (cSVF) (enzymatically derived from adipose tissue) in neuropathic trigeminal pain (n = 10). There was no control group. Preoperative NRS scores decreased 6 months after treatment [24].

#### 3.3.6. Neuropathic Pain without Any Apparent Cause

Beugels et al. (2018) studied the efficacy of AFG in non-neuroma neuropathic pain (n = 14). There was no control group. Thirteen patients were satisfied with the results 8 weeks after treatment, and eleven patients were satisfied at 48 months follow-up [15].

## 4. Discussion

This systematic review shows that AFG suppresses neuropathic pain but does not improve quality of life. Our results are based on no more than ten eligible clinical studies. The included clinical trials were heterogenous regarding the study population, sex, age, fat grafting indication, type of therapy, i.e., AFG, SVF, injection technique and study design.

AFG was the most studied indication for PMPS with pain reduction and improved quality of life in four out of five studies [17,18,20,21,24]. Nonetheless, a double-blinded RCT investigating rigottomy (subcisions for releasing and stretching scar tissue) with AFG compared to rigottomy with saline observed no differences in pain reduction, nor was the quality of life improved in PMPS. The scientific level of evidence in this study was low, for example, due to the small sample size that was used. Three other studies used a larger sample size but lacked a proper study design. In these studies, a placebo effect could not be ruled out because of the lack of a control group. Moreover, the results of the aforementioned studies were biased based on patient selection due to differences in baseline characteristics. For example, the studies included solely irradiated breast patients, who could be paired with radio-dystrophy, resulting in potentially smaller effects from AFG [18,19], patients prior to radiotherapy or chemotherapy [20,21], or patients who had all undergone reconstruction with subpectoral prothesis. Subpectoral prothesis placement could have caused pain in the area under investigation. Naturally, this could have influenced outcomes [17]. AFG (n = 9) and SVF-enriched fat grafting (n = 1) were studied as therapeutic components in the ten included studies.

The exact cause of neuropathic pain is still poorly understood, but neuropathic pain is mediated through several pathways. The peripheral de-sensitization of mechanoreceptive C fibers, for example, can result in pressure-induced pain or burning pain when C fibers fire spontaneously. Voltage-gated sodium and calcium channels strictly regulate the excitability of sensory neurons but cause enhanced neurotransmitter release and excitability. The firing of upregulated somatosensory neurons results in (chronic) neuropathic pain. Another possible cause of action is the abnormal expression of the transient receptor potential melastatin 8 channel, which leads to sensitization to cold and menthol responsiveness. Lastly, damaged nerves are in a pro-inflammatory state with the presence of M1-type macrophages (pro-inflammatory) and T cells. These immune cells release mediators, like tumor necrosis factor alpha and interleukins, that contribute to the sensitization of neurons through nerve destruction, thereby causing and maintaining neuropathic pain [2].

SVF-enriched fat grafting is thought to improve the regenerative potential of fat grafting because of the increased number of ASCs and the amount of ECM. ASCs produce anti-inflammatory paracrine factors, which are key in the treatment of neuropathic pain [25,26,27]. However, none of these studies tested the anti-inflammatory parameters before and after surgery. ASCs suppress T cell proliferation and the release of cytokines by CD4 (T helper) and CD8 (cytotoxic T) cells and stimulate the production of anti-inflammatory cytokine IL-10 by monocytes in vitro [28]. Other studies have confirmed the downregulation of inflammatory and T cell responses in vitro and in vivo [12,29,30]. In chronic nerve inflammation, nerves are exposed to a constant series of events of pro-inflammatory cytokine release that exacerbate neuronal injury [31]. The absence of repair leads to neuroma formation and altered nerve conduction, which is clinically expressed as neuropathic pain [32,33]. Hypothetically, downregulation would not only suppress the inflammatory situation around the nerves but it would also offer ASCs the ability to restore neuronal damage through immunomodulation, remyelination and axonal growth. One study administered ASCs intravenously in the caudal vein of a neuropathic pain mice model (sciatic nerve Chronic Constriction Injury—CCI). They found reduced levels of pro-inflammatory IL-1β, an activated and increased anti-inflammatory IL-10 and a restored nitric oxide synthase expression leading to reduced mechanical allodynia [34]. However, the intravenous injection of ASCs theoretically results in a short-lived effect due to the rapid diffusion of ASCs. The observed effect could, therefore, have been obscured or a temporary effect. A prolonged effect of ASCs is anticipated when SVF is administrated due to the presence of ECM. ECM functions as a slow-release scaffold for ASC paracrine factors and is able to guide cell differentiation and proliferation by binding cells [35]. Another animal study transplanted ASCs in sciatic nerve defects and compared the results to biodegradable, polycaprolactone-based nerve conduits without ASCs. The transplantation of ASCs resulted in the formation of a more robust nerve and a decrease in muscle atrophy [36]. Until now, however, the use of purified ASCs in humans has not been FDA-approved [37]. The FDA approves the clinical use of adipose tissue and tissue products when no significant risk to public health is expected. To meet these conditions, the product must be intended for homologous use only, and the production process may only be manipulated minimally. Therefore, the function of adipose tissue for providing the cushioning, support and storage of lipids has to be maintained. Autologous fat grafting and the extraction of stromal vascular fraction are considered to be minimal manipulation, whereas the isolation of adipose-derived stem cells from the use of enzymes is considered to be more than minimally manipulated. Currently, no adipose-derived cellular therapy is approved by the Center for Biologics Evaluation and Research (CBER) for the treatment of neuropathic pain. Other in vitro and in vivo studies mostly focus on the modification of bone marrow stromal cells or ASCs and their effects on nerve regeneration and inflammatory responses, with promising results [38,39,40,41]. 

The low scientific evidence of the included studies in terms of sample size, drop-outs, the use of control groups, risk of (selection) bias and lack of proper statistical analyses is limiting to make proper interpretation of this systematic review. Due to the low quality and heterogeneity in the included studies, it was not possible to perform a meta-analysis, which weakens the conclusions of this systematic review. The inclusion only human studies could be seen as either a strength or a limitation of this systematic review. Animal models are not easily translated to humans as animals heal faster, for example. Pain measurements are also difficult to perform and interpret in animals, even when it is feasible. On the other hand, the inclusion of animal studies could also contribute to a better understanding of the possible mechanism of AFG on neuropathic pain.

In future studies, AFG harvesting, injection techniques and dosing should be standardized, as well as the use of outcome measurements, the definition of neuropathic pain (duration, nature of pain) and follow-up time. Since nerve regeneration starts weeks after treatment and also continues for months, at least a 12 month follow-up is advised. Furthermore, the dose dependency and the repetition of treatment were not included in any of the studies as relevant influencing factors. For example, Sollie et al. injected lipoaspirate relative to the area of pain but did not standardize an injected volume per the affected surface area, which may form a confounding factor [22]. Future studies could consider investigating repetitive treatment as fat grafts may possibly reside in larger quantities around the neuroma and offer prolonged regenerative effects. Lastly, the use of concomitant procedures should be avoided in the context of a clinical trial. The study of Vickers et al. is prone to bias as one patient received facial botox injections during the time of the study, which could have affected the outcomes [24]. All these aforementioned recommendations for future clinical study designs are summarized in Table 6.

## 5. Conclusions

AFG is a new treatment for neuropathic pain and shows encouraging results in decreasing neuropathic pain. However, AFG did not improve patients’ quality of life. Current evidence for the therapeutic efficacy of AFG in the treatment of neuropathic pain is low based on limited eligible clinical studies. Further research should be conducted in RCTs, looking at the single or supplemented effect of AFG, SVF and ASCs on neuropathic pain. The implementation of standardized pain outcome measurements and standardized procedures should be met before definitive recommendations can be given.

## Figures and Tables

**Figure 1 bioengineering-11-00992-f001:**
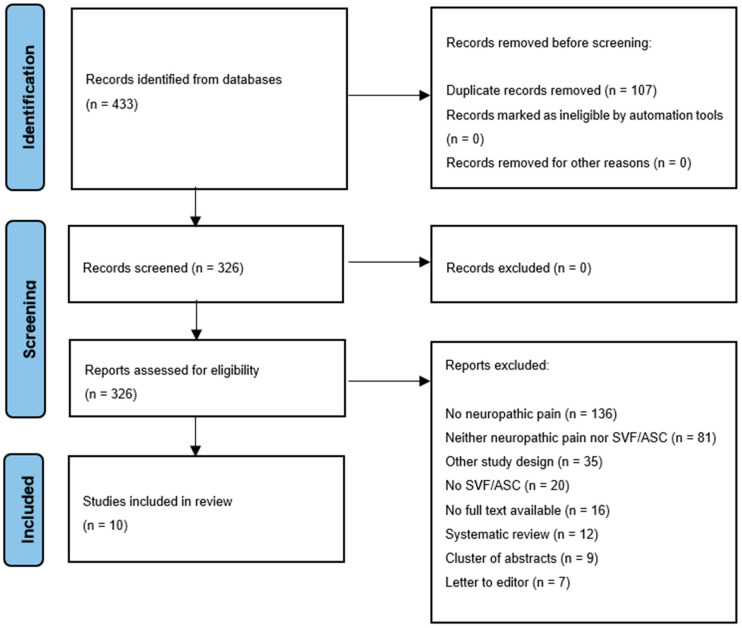
Flow diagram of study selection.

**Table 1 bioengineering-11-00992-t001:** Methodology Index for Non-Randomised Studies for all included studies.

	Beugels et al., 2018 [15]	Calcagni et al., 2016 [16]	Caviggioli et al., 2011 [17]	Caviggioli et al., 2016 [18]	Huang et al., 2015 [19]	Juhl et al., 2016 [20]	Lisa et al., 2020 [21]	Sollie et al., 2020 [22]	Sollie et al., 2022 [23]	Vickers et al., 2014 [24]
A clearly stated aim	2	2	2	2	2	2	2	2	2	2
Inclusion of consecutive patients	1	1	2	2	1	2	2	1	2	2
Prospective collection of data	1	1	2	2	1	2	2	1	2	2
Endpoints appropriate to the aim of the study	1	2	2	2	2	2	2	2	2	2
Unbiased assessment of the study endpoint	0	0	2	2	0	2	0	0	2	0
Follow-up period appropriate to the aim of the study	1	2	2	2	2	1	1	1	1	1
Loss to follow-up less than 5%	0	0	1	0	0	0	0	0	0	1
Prospective calculation of the study size	0	0	0	0	0	2	0	0	2	0
Only for comparative studies										
An adequate control group			1	1		2			2	
Contemporary groups			2	2		2			2	
Baseline equivalence of groups			1	1		1			1	
Adequate statistical analyses			1	1		2			2	
**Total MINORS score**	**6**	**8**	**18**	**17**	**8**	**20**	**9**	**7**	**20**	**10**
**Maximum possible score**	**16**	**16**	**24**	**24**	**16**	**24**	**16**	**16**	**24**	**16**

**Table 2 bioengineering-11-00992-t002:** Clinical trials assessing AFG in the treatment of neuropathic pain. RCS = Retrospective Cohort Study, PSS = Prospective single-center study. RCT = Randomized Controlled Trial, PMS = Prospective Multicenter Study, F = female, AFG = Autologous Fat Grafting, NP= Neuropathic Pain, QoL = Quality of Life, NRS = Numeric Rating Scale, VAS = Visual Analogue Scale, NPSI = Neuropathic Pain Symptom Inventory, POSAS = Patient and Observer Scar Assessment Scale, SF-36 = Short Form 36 (QoL outcome), cSVF = cellular Stromal Vascular Fraction, W = Weeks, M = Months.

Author (Year)	Study Design	N total (F)	Indication	Treatment	Age (Years)	Outcome	Follow Up (W/M)	Conclusion
Beugels et al., 2018 [15]	RCS	14 (8)	NP e.c.i	AFG	51 (17–78)	Tinel’s sign, VAS, patient satisfaction, quality of sleep.	2M 28M	AFG decreased treatment refractory NP. No conclusions can be drawn on patient-reported satisfaction and quality of sleep as they were not statistically tested.
Calcagni et al., 2016 [16]	RCS	5 (0)	Painful neuromas of radial nerve	Microsurgical resection SVF-enriched FG	49.8 ± 16.6	Subjective outcomes, overall pain score.	1M 6M 12M 36M	SVF-enriched FG did not reduce NP but tended to reduce NP over time.
Caviggioli et al., 2011 [17]	PSS	113 (113)	PMPS + scar retraction	AFG	NR	VAS Analgesic drug intake.	12M	AFG with scar retraction reduced NP. 28 out of 34 patients quit their analgesic drug therapy in the AFG treated group.
Caviggioli et al., 2016 [18]	PSS	209 (209)	PMPS	AFG	NR	VAS Analgesic drug intake.	12M	AFG reduced NP. 48 out of 120 patients quit their analgesic drug therapy in the AFG treated group.
Huang et al., 2015 [19]	PSS	13 (4)	Scar pain	AFG	33.1 ± 16.4	VAS, NPSI.	1W 1M 6M	AFG reduced NP.
Juhl et al., 2016 [20]	RCT	15 (15)	PMPS	AFG	T 58.9 ± 7.4 C 59.9 ± 9.8	DoloTest, VAS, NPSI, POSAS.	3M 6M	AFG reduced NP. Health-related QoL and scar quality improved.
Lisa et al., 2020 [21]	PMS	37 (37)	PMPS	AFG	48 ± 6.2	VAS, POSAS.	1M 3M 6M	AFG significantly reduced NP and improved scar quality.
Sollie et al., 2020 [22]	PSS	10 (NR)	Post-herpetic neuralgia	AFG	76 (53–94)	VAS, NPSI, SF-36.	3M	AFG reduced NP on all parameters. QoL did not improve (SF-36).
Sollie et al., 2022 [23]	RCT	35 (35)	PMPS	Scar release AFG	63.8 ± 9.9	NRS, NPSI, SF-36.	3M 6M	AFG did not reduce NP. QoL significantly improved in the control group and detoriated after AFG (on emotional parameters).
Vickers et al., 2014 [24]	PSS	10 (10)	Trigeminal pain	cSVF	55.3 ± 14.7	NRS Dosage of NP medication	6M	cSVF reduced NP. Dosage of NP medication tended to reduce over time.

**Table 3 bioengineering-11-00992-t003:** Methods of harvesting, processing, and injecting autologous fat. SD = standard deviation, NR = Not reported.

Author (Year)	Donor Site	Process	Injection Layer	Volume Injected
Beugels et al., 2018 [15]	Abdomen	NR	NR	NR
Caviggioli et al., 2011 [17]	Abdomen	Centrifugation 3000 rpm for 5 min.	Dermo-hypodermal junction	NR
Caviggioli et al., 2016 [18]	Abdomen	Centrifugation 3000 rpm for 5 min.	Dermo-hypodermal junction	52 ± 8.9 cc
Huang et al., 2015 [19]	Abdomen	Centrifugation 3000 rpm for 3 min.	Rigottomy Dermal-hypodermal junction	2.25 mL (range 0.3–14.0)
Juhl et al., 2016 [20]	Abdomen	Centrifugation 1800 rpm for 3 min.	Rigottomy Dermal-hypodermal junction	71 ± 24.6 mL
Lisa et al., 2020 [21]	Hips	Centrifugation 837× *g* for 5 min.	Dermal-hypodermal junction	NR
Sollie et al., 2020 [22]	Abdomen Thighs	Centrifugation 10 min. Speed NR.	Dermal-hypodermal junction	208 mL (range 100–300)
Sollie et al., 2022 [23]	NR	NR	Rigottomy Subdermal area	NR

**Table 4 bioengineering-11-00992-t004:** Methods of harvesting, laboratory processing and injecting technique of SVF-enriched fat.

Author (Year)	Donor Site	Laboratory Process	Injection Layer	Volume Injected
Calcagni et al., 2016 [16]	NR	Transferring lipoaspirate to the Celution 800/CRS System followed by adding one ampule of Celase enzyme reagent to isolate SVF.	Perineural area	A mixture of 5 mL concentrated SVF and 2 mL aspirated and sedimented lipid fraction.
Vickers et al., 2014 [24]	Bilateral lumbar region	Adding collagenase digestion of human grade collagenase to the lipoaspirate to isolate cSVF. Followed by the centrifugation in sterile saline-containing vancomycin.	Perineural area	NR

**Table 5 bioengineering-11-00992-t005:** As shown per clinical trial: its study population and corresponding results of the treatment of neuropathic pain. cSVF = cellular Stromal Vascular Fraction, VAS = Visual Analogue Scale, NPSI = Neuropathic Pain Symptom Inventory, POSAS = Patient and Observer Scar Assessment Scale, NRS= Numeric Rating Scale, T = Treatment, C = Control, OPS = Overall Pain Score, pain score from 0 to 3, NR = Not reported, NSD = No significant differences ↓ = decrease in score ↑ = increase in score.

Patients	Author (Year)	Intervention	Preoperative Outcome Measure	Postoperative Outcome Measure	*Δ* Pre- and Postoperative	Significance
14	Beugels et al., 2018 [15]	T: AFG in areas with demarcated NP (n = 14) C: N/A	T: VAS 7.4 (range 6.0–10.0)	T 2M: VAS 3.8 (range 0–8.0) T 6M: VAS 4.3 (range 0–10.0)	3.6 ↓ 3.1 ↓	*p* < 0.0001 *p* < 0.0017
5	Calcagni et al., 2016 [16]	T: Microsurgical resection + SVF-enriched FG (n = 5) C: N/A	T: Overall Pain Score (OPS) 2.2 ± 1.0	T 2M: OPS 1.3 ± 1.2 T 6M: OPS 1.5 ± 1.5 T 12M: OPS 1.6 ± 1.4 T 36M: OPS 1.4 ± 1.3	0.9 ↓ 0.7 ↓ 0.6 ↓ 0.8 ↓	*p* = 0.225 *p* = 0.345 *p* = 0.345 *p* = 0.104
113	Caviggioli et al., 2011 [17]	T: AFG in areas of NP + scars (n = 63) C: No intervention (n = 35)	T: VAS NR C: VAS NR	T 13M: VAS NR C 13M: VAS NR	3.2 ± 3.0 ↓ 1.0 ± 2.7 ↓	*p* = 0.0005 *p* = NR
190	Caviggioli et al., 2016 [18]	T: AFG in irradiated areas (n = 120) C: No intervention (n = 70)	T: VAS 7.2 ± 2.1 C: VAS 6.9 ± 2.2	T 12M: VAS 3.3 ± 3.1 C 12M: VAS 5.8 ± 1.9	3.2 ± 2.9 ↓ 1.1 ± 2.7 ↓	*p* < 0.005 *p* > 0.05
13	Huang et al., 2015 [19]	T: AFG in affected areas (n = 13) C: N/A	T: VAS 7.5 ± 1.1 NPSI 49.4 ± 13.3	T 1W: VAS 3.1 ± NR NPSI 25.0 ± 14.0	VAS 4.4 ± 1.7 ↓ NPSI 24.4 ± NR ↓	*p* = 0.009 VAS *p* = 0.004 NPSI
				T 1M: VAS 2.1 ± NR NPSI 21.0 ± 17.8	VAS 5.4 ± 2.1 ↓ NPSI 28.4 ± NR ↓	*p* = 0.008 VAS *p* = 0.0009 NPSI
				T 6M: VAS 1.9 ± NR NPSI 14.6 ± 16.9	VAS 5.6 ± 2.2 ↓ NPSI 34.8 ± NR ↓	*p* = 0.007 VAS *p* = 0.0008 NPSI
15	Juhl et al., 2016 [20]	T: AFG in painful areas (n= 8) C: Analgesic therapy (n= 7)	T: NR C: NR	T 3M: DoloTest NR VAS NR NPSI NR	DoloTest 220 ↓ (95% CI 146–294) VAS 36.0 ↓ (95% CI 27.1–44.9) NPSI 12.9 ↓ (95% CI 9.6–17.8)	*p* < 0.001 DoloTest *p* < 0.01 VAS *p* < 0.001 NPSI
				T 6M: DoloTest NR VAS NR NSPI NR	DoloTest 202 ↓ (95% CI 98–304) VAS 35.8 ↓ (95% CI 20.6–50.9) NPSI 13.7 ↓ (95% CI 9.6–17.8)	*p* < 0.001 DoloTest *p* < 0.01 VAS *p* < 0.001 NPSI
				C 3M: DoloTest NR VAS NR NPSI NR	DoloTest No change VAS No change NPSI No change	*p* = 0.36 DoloTest *p* = 0.43 VAS *p* = 0.55 NPSI
				C 6M: DoloTest NR VAS NR NPSI NR	DoloTest No change VAS No change NPSI No change	*p* = 0.25 DoloTest *p* = 0.28 VAS *p* = 0.68 NPSI
37	Lisa et al., 2020 [21]	T: AFG in area of NP (n = 37) C: N/A	T: VAS 6.9 ± 1.3	T 1M: VAS 3.8 ± 1.6 T 3M: VAS 3.0 ± 1.6 T 6M: VAS 2.6 ± 2.1	VAS 3.1 ± NR ↓ VAS 3.9 ± NR ↓ VAS 4.3 ± NR ↓	*p* < 0.001 *p* < 0.005 *p =* NR
10	Sollie et al., 2020 [22]	T: AFG to dermal area of neuralgia (n = 10) C: N/A	T: VAS average 7.0 ± 1.6 VAS max. 8.6 ± 1.3 NPSI average 5.8–7.2 ± 1.8–3.2 SF-36 pain 38.8 ± 28.4	T 3M: VAS average 3.0 ± 3.0 VAS max. 3.5 ± 3.1 NPSI average 1.9–3.0 ± 2.2–3.1 SF-36 pain 49.8 ± 26.4	VAS average 4.0 ± 3.1 ↓ VAS max. 5.1 ± 3.9 ↓ NPSI average 3.2–4.5 ± 3.1–4.9 ↓ SF-36 pain 11.0 ± 39.7 ↑	*p* < 0.05 VAS *p* < 0.05 VAS *p* < 0.05 NPSI average *p* = 0.40 SF-36 pain
35	Sollie et al., 2022 [23]	T: Scar releasing rigottomy + AFG (n = 18) C: Scar releasing rigottomy + saline injection (n = 17)	T: NRS average 5.3 (95% CI 4.6–6.0) NRS max 6.6 (95% CI 5.9–7.4) NPSI average calculated 3.3 (95% CI) SF-36 pain 53.9 (95% CI 45.3–62.4) C: NRS average 5.3 (95% CI 4.4–6.2) NRS max 7.1 (95% CI 6.3–8.0) NPSI average calculated 4.1 (CI 95%) SF-36 pain 49.1 (CI 95% 42.6–55.7)	T 3M: NRS average 4.1 (95% CI 2.9–5.3) NRS max 5.8 (95% CI 4.8–6.8) NPSI average calculated 2.8 (95% CI) SF-36 pain NR	NRS average 1.2 ± NR ↓ NRS max 0.8 ± NR ↓ NPSI average calculated 0.7 ↑ SF-36 pain NR	*p* = NSD for NRS, NPSI and SF-36 after 3M and 6M compared to baseline
				T 6M: NRS average 4.3 (95% CI 3.1–5.4) NRS max 5.7 (95% CI 4.4–7.0) NPSI average calculated 2.7 (95% CI) SF-36 pain 57.1 (46.6–67.5)	NRS average 1.0 ± NR ↓ NRS max 0.9 ± NR ↓ NPSI average range 2.0–0.2 ↓ SF-36 pain 3.2 ↓	
				C 3M: NRS average 4.7 (95% CI 3.5–5.9) NRS max 6.3 (95% CI 5.0–7.5) NPSI NR SF-36 pain NR	NRS average 0.6 ± NR ↓ NRS max 0.8 ± NR ↓ NPSI NR SF-36 NR	
				C 6M: NRS average 4.3 (95% CI 3.1–5.4) NRS max 5.9 (95% CI 4.6–7.2) NPSI NR SF-36 pain 52.1 (95% CI 42.1–62.0)	NRS 1.0 ± NR ↓ NRS 1.2 ± NR ↓ NPSI NR SF-36 pain NR	
10	Vickers et al., 2014 [24]	T: cSVF in facial NP (n = 10) C: N/A	T: NRS 7.5 ± 1.6	T 6M: NRS 4.3 ± 3.3	3.2 ± NR ↓	*p* = 0.018

**Table 6 bioengineering-11-00992-t006:** Recommendations for future clinical studies. * VAS = Visual Analogue Scale, ** NPSI = Neuropathic Pain Symptom Inventory.

Topic	Recommendations
**Study quality**	1. Controlled (comparison with gold standard or placebo) 2. Randomized 3. Follow up 12 months 4. Statistical testing for differences between groups
**Procedure standardization**	1. Standardize AFG, SVF or ASCs harvesting, processing and injection technique 2. Document and correct for fat specific characteristic such as gender, BMI, smoking and co-morbidities e.g., diabetes 3. Standardize injected volume per surface area. In case of supplemented-AFG: standardized volume-to volume ratio 4. Avoid concomitant procedures or correct for them
**Pain measurements **	1. Define pain and pain reduction. Standardize follow-up moments 2. Use valid pain outcome measures (VAS *, NPSI **) 3. Report on all possible bias on pain experience e.g., use of analgesia, pain blocking procedures, psychological factors

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
