# Peer review of "Therapeutic Efficacy of Adipose Tissue-Derived Components in Neuropathic Pain: A Systematic Review"

_bioengineering, 2024, doi:10.3390/bioengineering11100992_

Round 1
Reviewer 1 Report
Comments and Suggestions for Authors
The topic is interesting but, despite the laudable efforts of the Authors, the result is very poor. The articles identified are only 10 and moreover they are not very homogeneous. These are 4 disparate case reports and 6 cohorts related to the treatment of scars : of the latter, 5 are all related to mastectomy sequelae, of which 3 come from the same working group. Moreover, the Authors themselves admit a priori low scientific evidence of the referred studies.
I believe that there should have been a Neurologist in the group of authors.
The possible mechanism of pain onset should have been discussed in the different clinical situations presented and therefore the possible different biological role that the fat graft could have had in such situations should have been discussed. For example, considering that the effectiveness of fat grafting for the release of scar tissue has been demonstrated, the improvement in pain could simply be related to the lower pressure exerted on the sensory endings.
Actually, the Authors mention the production of antinflammatory paracrine factors by the fat graft, but this has not been tested in the referred studies.
I would like to point out that laws and regulations of the therapeutic use of ASCs were reported, but not presented and I think that the regulatory aspect of this procedure is of considerable importance.
As a small imperfection, in the key word the Authors mention allogenic fat graft, but I was not able to find any trace of this particular type of graft in the paper.
Author Response
Comment: I believe that there should have been a Neurologist in the group of authors.
Answer: Thank you for your review. A neurologist, and perhaps also an anesthesiologist, could be an
addition to the team of authors for sharing their professional opinion and experiences in the treatment
of neuropathic pain. However, this article is an overview of current clinical studies about neuropathic
pain after AFG or adipose derived components, where inclusion of these additional authors will not
lead to different outcomes. In our opinion the knowledge and experience of prof. dr. M.C. Harmsen as
a professor in the field of regenerative medicine and stem cell biology, and prof. dr. J.H. Coert as
plastic surgeon with expertise in peripheral nerve surgery, are of important value for this review and
more specific its discussion.
Comment: The possible mechanism of pain onset should have been discussed in the different clinical
situations presented and therefore the possible different biological role that the fat graft could have
had in such situations should have been discussed. For example, considering that the effectiveness
of fat grafting for the release of scar tissue has been demonstrated, the improvement in pain could
simply be related to the lower pressure exerted on the sensory endings. Actually, the Authors mention
the production of antinflammatory paracrine factors by the fat graft, but this has not been tested in the
referred studies.
Answer: Possible mechanisms of action in neuropathic pain are described in our discussion, (line
189-200). The mechanism of action for SVF enriched fat is assigned to potential regenerative
qualities (line 201-203). ASCs are already known for their immunomodulatory benefits in the
treatment of neuropathic pain. Unfortunately, the included clinical studies did not include any
regenerative or anti-inflammatory effects in their outcome measures. This is widely discussed in the
discussion. In vitro and in vivo studies have confirmed these immune modulatory effects, these are
well described in our discussion (line 205-229). Until now, the use of purified ASCs in humans is not
FDA approved.
Comment: would like to point out that laws and regulations of the therapeutic use of ASCs were
reported, but not presented and I think that the regulatory aspect of this procedure is of considerable
importance.
Answer: Thank you for your comment. We have added a paragraph about the regulatory process of
the FDA. See changes in red.
Comment: As a small imperfection, in the key word the Authors mention allogenic fat graft, but I was
not able to find any trace of this particular type of graft in the paper.
Answer: If we are correct, our keywords do not include allogenic fat graft. It was included in our
systematic literature search, however, not included in our paper.
Reviewer 2 Report
Comments and Suggestions for Authors
It is a systematic and comprehensive review of the existing bibliography on adipose tissue derivatives to treat neuropathic pain. The review proposes some directions for future clinical research in this field. It highlights the lack of standardization in harvesting, processing, and injecting techniques for adipose tissue-derived therapies. The review mentions the potential anti-inflammatory effects of adipose tissue-derived components. However, no one of the included studies specifically tested these parameters. The comorbidity between obesity and neuropathic pain could also be discussed.
Author Response
Comment: It is a systematic and comprehensive review of the existing bibliography on adipose tissue
derivatives to treat neuropathic pain. The review proposes some directions for future clinical research
in this field. It highlights the lack of standardization in harvesting, processing, and injecting techniques
for adipose tissue-derived therapies. The review mentions the potential anti-inflammatory effects of
adipose tissue-derived components. However, no one of the included studies specifically tested these
parameters.
Answer: Thank you for your comment. This answer is an addition to our answer to the previous
reviewer concerning this topic. Indeed, none of the studies tested the anti-inflammatory effects.
However, adequate references were found to substantiate these potential anti-inflammatory effects
and in our opinion they are important to mention in the introduction. Therefore, no adaptation was
made to the introduction. The discussion already underlines the fact that none of the included studies
have tested these potential beneficial effects, see line (line 201-204).
Comment: The comorbidity between obesity and neuropathic pain could also be discussed.
Answer: The relation between obesity and the incidence of pain, and more specific neuropathic pain,
is an interesting topic. We have chosen not to include this topic including its potential cellular
mechanism in our already extensive discussion. The limited studies and the small study populations
are not large enough to distinguish between obesity and its adverse effect on neuropathic pain.
Reviewer 3 Report
Comments and Suggestions for Authors
This is a systematic review on the "efficacy of adipose-derived components in neuropathic pain". The manuscript is well-structured, in adaequate English and the topic is certainly clinically relevant and of significant interest to the audience of Bioengineering and especially this special issue.
The main merit in my opinion of this paper is that it shows the low evidence of the only 10 relevant studies included to support fat grafting to reduce nerve-related pain. This sobering finding surprised me and may influence also the clinical decision-making of other readers.
In summary, I would recommend to accept the manuscript for publication after minor revision.
However, I would suggest the following adaptions:
Abstract:
Line 25: neurop-thy - no -
Abstract adaequat, yet I would state more clearly in the conclusion that the current evidence for therapeutic efficacy of using autologous adipose tissue to reduce neuropathic pain is very low.
Introduction:
Line 65, references 10, 12, 13 - do they investigate the effect of adipose tissue in neuropathic pain ? maybe add a reference that is specifically targeted at neuropathic pain ?
Methods: seems correct according to PRISMA requirements
Line 73 ... Were "painful scars" also included ? I miss the search term and for example the indication of fat grafting in painful neuroma after episiotomy (e.g. Ulrich et al. Int J Gynaecol Obstet 2011)
Results: Figure 1: Would it be possible to enhance the letter size ?
Line 111: Why was the period between 2011 and 2022 chosen specifically ?
Table 1, 2 and 3: Please enhance letter size - not readable without loops !
Discussion: well-structured, precise and inspiring !
Line 176 - Please clarify the term "rigottomy" which may be unfamiliar to the average reader or better use another term
Table 5: completely unreadable, please revise !
Table 6: Enhance letter size, please !
Conclusion: Section seems very short, please expand !
Author Response
This is a systematic review on the "efficacy of adipose-derived components in neuropathic pain". The
manuscript is well-structured, in adaequate English and the topic is certainly clinically relevant and of
significant interest to the audience of Bioengineering and especially this special issue.
The main merit in my opinion of this paper is that it shows the low evidence of the only 10 relevant
studies included to support fat grafting to reduce nerve-related pain. This sobering finding surprised
me and may influence also the clinical decision-making of other readers.
In summary, I would recommend to accept the manuscript for publication after minor revision.
However, I would suggest the following adaptions:
Comment: Abstract: Line 25: neurop-thy - no –.
Answer: Adapted.
Comment: Abstract adaequat, yet I would state more clearly in the conclusion that the current
evidence for therapeutic efficacy of using autologous adipose tissue to reduce neuropathic pain is
very low.
Answer: Thank you for your comment. We emphasized the current limited evidence in the conclusion
of the review.
Comment: Line 65, references 10, 12, 13 - do they investigate the effect of adipose tissue in
neuropathic pain ? maybe add a reference that is specifically targeted at neuropathic pain ?
Answer:
‘ The clinical application of adipose tissue has been mostly described in chronic wounds and soft
tissue reconstructions. 11 Few is known about the role of adipose tissue in the treatment of neuropathic
pain. 10,12,13 This systematic review therefore aims to investigate the therapeutic clinical efficacy of
adipose tissue-derived components in neuropathic pain. ‘
No, these studies did not investigate the effect of adipose tissue on neuropathic pain. The first study
is about the procedure to isolate SVF intraoperative versus nonintraoperative (10). The second study
describes potentials and limitations of bone marrow stroma in tissue engineering and the isolation
process of stem cells from human lipoaspirate. (12) The third study establishes definitions of stromal
cells as uncultured SVF and as stem cells. (13)
The purpose of this review was to find studies assessing the role of adipose tissue in the treatment of
neuropathic pain. Therefore, no extra reference was added.
Methods: seems correct according to PRISMA requirements
Comment: Results: Figure 1: Would it be possible to enhance the letter size ?
Answer: We agree. The editorial office will perform changes to the layout of the tables once the
article is accepted for publication.
Comment: Line 111: Why was the period between 2011 and 2022 chosen specifically ?
Answer: This period was not chosen. The literature search was performed from inception to August
2023 where the period of 2011 to 2022 was based on results from our literature search.
Comment: Table 1, 2 and 3: Please enhance letter size - not readable without loops !
Answer: We agree. The editorial office will perform changes to the layout of the tables once the
article is accepted for publication.
Comment: Discussion: well-structured, precise and inspiring!
Answer: Thank you for your comment!
Comment: Line 176 - Please clarify the term "rigottomy" which may be unfamiliar to the average
reader or better use another term
Answer: We elucidated the term "rigottomy" with a short explanation to be more clear to the readers.
Changes can be found in red.
Comment: Table 5: completely unreadable, please revise !
Answer: We agree. The editorial office will perform changes to the layout of the tables once the
article is accepted for publication.
Comment: Table 6: Enhance letter size, please !
Answer: We agree. The editorial office will perform changes to the layout of the tables once the
article is accepted for publication.
Comment: Conclusion: Section seems very short, please expand !
Answer: We extended our conclusion to emphasize the low evidence for AFG in the treatment of
neuropathic pain based on the limited available clinical studies. See red changes.
Reviewer 4 Report
Comments and Suggestions for Authors
Dear Authors, thanks for submitting your paper to this journal. The Authors have made a valuable systematic review of the therapeutic efficacy of adipose tissue-derived components in neuropathic pain. The review is interesting, clearly written and the methodology is well structured. They have made a comprehensive literature search and presented the findings in a clear manner. Anyway, I believe that the discussion needs revision. I think that discussing the findings of in vivo and in vitro studies (excluded from the review) is important but probably it would fit better in the introduction section. Moreover, it will be interesting to discuss more the results of the reviewed articles.
I noticed some typing errors:
Line 197: the is an extra full stop.
Line 25: neuropa-thy
Author Response
Dear Authors, thanks for submitting your paper to this journal. The Authors have made a valuable
systematic review of the therapeutic efficacy of adipose tissue-derived components in neuropathic
pain. The review is interesting, clearly written and the methodology is well structured. They have
made a comprehensive literature search and presented the findings in a clear manner. Anyway, I
believe that the discussion needs revision. I think that discussing the findings of in vivo and in vitro
studies (excluded from the review) is important but probably it would fit better in the introduction
section. Moreover, it will be interesting to discuss more the results of the reviewed articles.
I noticed some typing errors:
Comment: Line 197: the is an extra full stop.
Answer: Removed.
Comment: Line 25: neuropa-thy
Answer: Adapted.
Round 2
Reviewer 3 Report
Comments and Suggestions for Authors
The manuscript has been revised in accordance / reply to the reviewer`s suggestions and I would recommend its acceptance for publication.